# Classification and Prognosis Analysis of Pancreatic Cancer Based on DNA Methylation Profile and Clinical Information

**DOI:** 10.3390/genes13101913

**Published:** 2022-10-21

**Authors:** Xin Li, Xuan Zhang, Xiangyu Lin, Liting Cai, Yan Wang, Zhiqiang Chang

**Affiliations:** 1Harbin Medical University Cancer Hospital, Harbin 150081, China; 2Harbin Institute of Technology, School of Life Science and Technology, Harbin 150001, China; 3The First Affiliated Hospital of Baotou Medical College Cancer Center, Baotou 014016, China; 4College of Bioinformatics Science and Technology, Harbin Medical University, Harbin 150081, China

**Keywords:** pancreatic adenocarcinoma, DNA methylation, prognosis, consistency clustering, tumor classification

## Abstract

Pancreatic adenocarcinoma (PAAD) has a poor prognosis with high individual variation in the treatment response among patients; however, there is no standard molecular typing method for PAAD prognosis in clinical practice. We analyzed DNA methylation data from The Cancer Genome Atlas database, which identified 1235 differentially methylated DNA genes between PAAD and adjacent tissue samples. Among these, 78 methylation markers independently affecting PAAD prognosis were identified after adjusting for significant clinical factors. Based on these genes, two subtypes of PAAD were identified through consistent clustering. Fourteen specifically methylated genes were further identified to be associated with survival. Further analyses of the transcriptome data identified 301 differentially expressed cancer driver genes between the two PAAD subtypes and the degree of immune cell infiltration differed significantly between the subtypes. The 14 specific genes characterizing the unique methylation patterns of the subtypes were used to construct a Bayesian network-based prognostic prediction model for typing that showed good predictive value (area under the curve value of 0.937). This study provides new insight into the heterogeneity of pancreatic tumors from an epigenetic perspective, offering new strategies and targets for personalized treatment plan evaluation and precision medicine for patients with PAAD.

## 1. Introduction

Pancreatic adenocarcinoma (PAAD) is a malignant tumor of the digestive tract. New cases accounted for 2.6% and 4.7% of all new cancers and cancer-related deaths worldwide in 2020, respectively, and PAAD is the seventh leading cause of cancer deaths [1]. Furthermore, PAAD has been projected to overtake breast cancer as the third leading cause of cancer-related deaths by 2025 [2]. Although targeted and immunotherapy agents have been utilized in PAAD treatment, its overall 5-year survival rate remains <10%.

Clinical features such as tumor-modal-metastasis staging are commonly used for clinical evaluation and prognosis analysis of PAAD. However, research on the molecular classification of PAAD is still in its infancy, and thus there is no standard classification system available for patient stratification. This is further complicated by the heterogeneity of PAAD, which contributes to the dearth of molecular classifications for different sensitivities that could be widely used clinically for prognostic assessment and treatment guidance [3].

Therefore, it is necessary to develop more effective methods for the clinical diagnosis and prognosis assessment of PAAD, particularly for early detection, diagnosis, and treatment. Studies have confirmed that the occurrence and development of tumors are closely related to the activation of oncogenes and inactivation of tumor suppressor genes [4,5], which involve cumulative and progressive genetic changes. Epigenetic modification (represented by DNA methylation) is also considered one of the important factors related to the etiology of malignant tumors [6].

Abnormal methylation of the gene promoter region causes tumor suppressor gene silencing, which may further promotes tumorigenesis and development [7]. In addition, tumors originating from different tissues or organs show strong genetic and epigenetic heterogeneity. Although the pathogenesis of tumors is not entirely clear, increasing evidence supports an association with aberrant methylation [8,9]. Furthermore, the study of DNA methylation is useful for analyzing the epigenetic heterogeneity of tumors, and its status is a potential molecular marker of cancer subtypes [8,9].

The accumulation of high-throughput sequencing molecular data provides an opportunity to explore the specific molecular characteristics of unique potential subtypes of PAAD and define clinically applicable molecular disease subgroups [3]. Moreover, machine-learning algorithms and statistical methods are constantly being updated, providing new research approaches for analyzing high-throughput sequencing and clinical data.

These data can be further used to construct prognostic prediction models for patients with PAAD. For example, some studies have used the least absolute shrinkage and selection operator (lasso) [10,11] method to identify potential molecular features in large sample data. For PAAD, molecular data and clinical information from patients could be used as potential prognostic factors. Previous studies on other tumors have generally integrated various machine learning algorithms [12,13,14] to obtain the optimal model.

This study aims at mining molecular markers of DNA methylation in PAAD and identifying DNA methylation subtypes related to prognosis. Bioinformatics algorithms are used to explore the specific molecular markers and prognosis among subtypes of PAAD. Based on the DNA methylation data of PAAD, 831 significant high differential methylation genes and 404 significant low differential methylation genes were identified between PAAD and adjacent samples. Using the COX proportional hazard regression model and introducing significant clinical factors affecting prognosis, 78 genes were identified. The methylation level of these genes is an independent factor affecting the prognosis of PAAD. Based on the unsupervised clustering analysis of the selected prognosis-related genes, two molecular subtypes with significant differences in prognosis were obtained by consistent clustering algorithm, and 14 specific methylation genes were further identified. It can represent the unique methylation patterns of these two subtypes. Our findings provide new ideas and targets for individualized treatment and prognosis assessment of patients with PAAD.

## 2. Materials and Methods

### 2.1. Acquisition of DNA Methylation Data and Clinical Information of Patients with PAAD

The DNA methylation and expression data of 184 samples of patients with PAAD from The Cancer Genome Atlas (TCGA) database and their corresponding clinical information were collected. All methylation data were generated using the Illumina Infinium Human Methylation 450 BeadChip (HM450K chip) platform. The expression information in this dataset and the DNA methylation data used in this study were matched using SampleID to further explore the heterogeneity of PAAD subtypes. The clinical data included sex, age, stage, overall survival (OS) status, and OS time to identify clinical factors that significantly affect patient survival.

The data were preprocessed to remove methylation sites with missing methylation level (β) values exceeding 70% of the total number of samples, and the k-nearest neighbor method was used to supplement the null values to further standardize the data. Probe sites belonging to the promoter region in cancer sample data were selected by defining the promoter region as 2 kb upstream to 0.5 kb downstream of the transcription start site. Loci corresponding to single-nucleotide polymorphisms (SNPs), the sex chromosomes, and more than one gene were removed. Because the HM450K chip detects the genome-wide methylation level of more than 480,000 CpG loci, some genes with multiple CpG loci will be detected. Therefore, we calculated the mean methylation level of CpG sites within each gene as the gene promoter region methylation level, and this value was used in subsequent analyses. The 450K chip data of PAAD methylation obtained from TCGA database contained 485,577 CpG probe sites. The CpG sites with excessive missing values were removed, the missing values of the remaining CpG sites were adjusted, the sites corresponding to sex chromosomes and SNPs were removed, and CpGs located in the promoter region were extracted. After these preprocessing steps, 131,288 CpG site methylation values of each sample were obtained. The methylation levels of these CpG sites were then used to characterize the corresponding 16,420 genes and the methylation profiles of PAAD genes were obtained (Appendix A). GeneSymbol represents a unique coding name for each probe site, with methylation values (β) ranging from 0 to 1, corresponding to an unmethylated and fully methylated state, respectively.

The expression data of PAAD were obtained from integrated TCGA transcriptome data based on the IlluminaHiSeq 2000 sequencing platform. These data were standardized by log2 (XML1) and the genes were mapped to human genome coordinates by UCSC Xena Hugo ProbeMap. The expression information in this data set was matched with the DNA methylation data according to the unique Sample ID.

### 2.2. Screening and Analysis of Differentially Methylated Genes

R software was used to perform enrichment analysis and functional annotation for 1235 differentially methylated DNA genes based on the Gene Ontology (GO) [15] and Kyoto Encyclopedia of Genes and Genomes (KEGG) databases [16]. Their enrichment analysis was carried out based on the hypergeometric test, and the *p*-values were calculated as follows:(1)$$E-S_{GO}\left(S,g_{j}\right)=-log_{10}\left(\overset{n}{\underset{k=m}{\sum }}\frac{\left(\begin{array}{c} M\\ k \end{array}\right)\left(\begin{array}{c} N-M\\ n-k \end{array}\right)}{\left(\begin{array}{c} N\\ n \end{array}\right)}\right)$$
where *N* is the total number of human proteins; *M* is the number of proteins annotated as $g_{j}$; *n* is the number of proteins in G(c); and m represents proteins in G(c), annotated as the number of $g_{j}$.
(2)$$E-S_{KEGG}\left(c,P_{j}\right)=-log_{10}\left(\overset{n}{\underset{k=m}{\sum }}\frac{\left(\begin{array}{c} M\\ k \end{array}\right)\left(\begin{array}{c} N-M\\ n-k \end{array}\right)}{\left(\begin{array}{c} N\\ n \end{array}\right)}\right)$$
where *N* is the total number of human proteins, M is the amount of protein annotated as $P_{j}$ of the KEGG pathway, *n* is the number of proteins in G(c), and m is the number of proteins in G(c) and $P_{j}$.

### 2.3. Identification of Methylation Markers Associated with Prognosis

To identify the molecular subtypes of PAAD related to prognosis, 184 samples obtained from TCGA database were divided into training and test sets (128 and 56 samples, respectively) according to a 7:3 ratio. OS time and survival status (0 representing survival and 1 representing death) were used to construct survival events, which were employed in Kaplan–Meier survival analysis of the two methylated subgroups using the survival functions *Survfit* and *Survdiff* in the R package. In univariate analysis, the model only describes the relationship between each single variable and survival status, ignoring the influence of other variables; however, the survival status associated with a given sample/patient may also be affected by different clinical factors. To address this problem, the univariate Cox proportional hazards regression model was used to include the significant prognostic genes (*p* < 0.05) into the multivariate Cox proportional hazards regression model, and the significant prognostic clinical factor age (*p* = 0.021) was used as a covariate to find independent prognostic factors. For each feature *i*, the univariate and multivariate Cox proportional hazards regression models are described in Equations (3) and (4), respectively:(3)$$h{\left(t,x\right)}_{i}=h0\left(t\right)exp\left(\beta_{methy}methy_{i}\right)$$
(4)$$h{\left(t,x\right)}_{i}=h0\left(t\right)exp\left(\beta_{methy}methy_{i}+\beta_{age}age\right)$$
where *h*0 (*t*) is the benchmark risk equation, which can be any non-negative equation for time t; $methy_{i}$ is the vector representing the methylation level of $CpG_{i}$; age is the vector representing the clinical stage of the patient; and $\beta_{methy}$ and $\beta_{age}$ represent the regression coefficients. A positive regression coefficient indicates that the increase of methylation level is related to an increased risk of death, whereas a negative coefficient indicates that the increase of methylation level is related to a decreased risk of death. The Benjamini–Hochberg method was used to adjust the *p*-value.

After univariate and multivariate Cox proportional hazard regression analysis, the significant independent prognostic markers were used to construct a risk score (Risk Score) in the training set, represented by a linear combination of the DNA methylation levels of these markers and corresponding regression coefficients, as follows:(5)$$Risk\;Score=\sum_{i=1}^{n}\beta_{i}X_{i}$$
where $\beta_{i}$ is the Cox regression coefficient of gene *i* in the training set, $X_{i}$ is the DNA methylation level of gene *i*, and *n* is the number of genes that significantly affect survival.

The training set was used for model construction and screening of prognostic markers, and the test set was used for subsequent verification of the selected prognostic markers.

### 2.4. Subtype Recognition

To identify different subtypes of PAAD based on CpG locus data with significant prognostic correlations to survival, unsupervised consistent clustering analysis was performed using the ConcensusClusterPlus R package [17]. We tested multiple clustering algorithms and metric distances embedded in the ConcensusClusterPlus package to explore the optimal clustering results. Various clustering algorithms were applied to the dataset, with the K-means clustering algorithm using the Euclidean distance as a metric and the PAM clustering algorithm using the Pearson distance as the metrics finally considered. The cumulative distribution function of consistency, mean coefficient of consistency, and coefficient of variation (CV) were then used to determine the final number of subtypes. The final criteria used to determine the clustering method and the number of clusters were: certain K cluster numbers, the consistency value between groups was relatively high, and the CV was relatively low. The CV was calculated as follows:CV = (SD/MN) × 100%(6)
where SD is the standard deviation of the consistency value among different groups under the same number of categories, and MN is the average consistency among different groups under the same number of categories. Patients with PAAD were classified into two subtypes based on the DNA methylation levels of the mined prognosis-related genes.

### 2.5. Identification of Subtype Specific Marker Genes

The average methylation level of each gene in Cluster1 and Cluster2 subtype samples was calculated using the custom R script identification. The mean difference was calculated and the significance of the difference was evaluated by a two-sided t-test. Genes with a mean β difference greater than 0.1 and *p* value less than 0.05 after multiple test correction by the Benjamini-Hochberg method were considered to be differentially methylated genes. Next, QDMR [18] was used to further screen the sites with high or low methylation specificity in PAAD subgroups.

### 2.6. Construction of a Prognosis Prediction Model for PAAD Typing

The DNA methylation subtypes identified in this study were obtained by the aforementioned unsupervised consistent clustering method. To validate the DNA methylation typing results and to establish a more convenient and accurate methylation typing method for PAAD, a supervised Bayesian network classification model was constructed using the training set. In a Bayesian network, the root node is represented by nodes indicating clusters (i.e., Cluster1 and Cluster2), and all other feature nodes are used as “offspring” extending from the root node. The Bayesian network classifier is used to calculate the probability of a sample belonging to a category and then determine which category it belongs to, rather than directly assigning a category to a given sample. With this classifier, we used the 14 specific genes obtained in the previous step as features and the two DNA methylation subtypes as classification labels. Ten-fold cross-validation was used during model training to better evaluate the performance of the model. Feature curves (receiver operating characteristic curves) were constructed using the pROC package in R software.

### 2.7. Expression Differences According to DNA Methylation Subtype

Expression data of TCGA PAAD samples were matched with previously obtained related DNA methylation data using Sample ID. A custom R script was used to identify differentially expressed genes among DNA methylation subtypes based on gene expression levels. Differentially expressed genes were defined as follows: the mean ratio (fold change) of expression levels of genes between the two groups of samples was >2 or <0.5, and the *p* value of the paired *t*-test was <0.05 after correction by the Benjamini–Hochberg method.

### 2.8. Analysis of Immune Cell Abundance in DNA Methylation Subtypes

Studies have shown that immune cell dysfunctions, such as abnormal distribution of abundance and type, as well as developmental and functional abnormalities are associated with various diseases, including cancer [19]. Therefore, studying the extent of immune cell infiltration can provide important assessments of immune status, disease progression and prognosis, and treatment. In this study, the ImmuCellAI algorithm [20] was used to calculate and evaluate the abundance of 24 types of immune cells in PAAD samples based on RNA expression profiles, including 18 T cell subsets [CD4+, CD8+, CD4+ naive, CD8+ naive, central memory T, effector memory T, Tr1, induced regulatory T cells (iTreg), natural regulatory T cells (nTregs), Th1, Th2, Th17, Tfh, Tc, mucosal-associated invariant T (MAIT), Tex, γ-delta T, and natural killer (NK) T cells] and six other important immune cells (B cells, macrophages, monocytes, neutrophils, dendritic cells, and natural killerNK cells). Evaluation of the degree of immune cell infiltration can be used as a prognostic factor and a predictor of treatment efficacy [21], which has strong guiding significance for selecting patients to receive immunotherapy.

## 3. Results

### 3.1. Identification of Methylation Markers Associated with Prognosis in PAAD

We identified 1235 differentially methylated genes between cancer and para-cancer samples among 16,420 genes with methylation information on PAAD, including 831 and 40 differentially hypermethylated and hypomethylated genes, respectively. The visualization of the distribution of these differentially methylated genes is presented as a volcano diagram constructed using the R package GGPlot (Figure 1A). The 10 most significantly differentially hypermethylated genes were *SARM1*, *IRX4*, *ZSCAN23*, *FOXC2*, *PTPN5*, *IRF4*, *CACNA1*, *EN2*, *HOXB4*, and *IGF2BP1,* and the 10 most significantly differentially hypomethylated genes were *REG4*, *C11orf34*, *BRD9*, *S100A16*, *HIST1H2BK*, *STATH*, *LRRC31*, *UBD*, *MIR548A1*, and *PSMG3*.

GO and KEGG pathway enrichment analyses were, respectively, performed on the 1235 differentially methylated genes (Figure 1B,C). In the GO functional enrichment set (Figure 1B), the most significant enrichment functions mainly included the neuropeptide signaling pathway, neuronal differentiation in the central nervous system (CNS), embryonic organ development, and other biological processes. The KEGG pathway enrichment analysis showed that these genes were mainly involved in neuroactive ligand receptor interaction, calcium, cAMP, circadian rhythm, morphine addiction, glutamatergic synapses, pancreatic secretions, salivary secretion, and nicotine addiction signaling pathways (Figure 1C).

To identify molecular subtypes of PAAD associated with prognosis, genes significantly affecting survival were included as taxonomic features. The univariate Cox proportional hazards regression model was constructed using the 1235 differentially methylated genes and three clinical factors identified for the samples in the training set. Finally, 135 genes were found to be significantly associated with the survival of patients with PAAD (*p* < 0.05). In addition, the univariate Cox proportional hazards regression model established for clinical factors (Appendix A) showed that “age” (*p* = 0.021) was also a significant factor affecting the survival of patients. The multivariate Cox proportional hazards regression model included these 135 significantly different methylated genes, along with age as a covariate to identify factors that independently affect prognosis. Finally, 78 methylated genes significantly associated with prognosis were identified using multivariate Cox regression models (Appendix A).

### 3.2. Construction of a Prognosis Prediction Model for PAAD Classification

The methylation values (β) of 78 methylated genes independently associated with the prognosis of PAAD obtained in this study were consistently clustered. Analysis of the clustering methods used revealed that the PAM (Figure 2A) and K-means (Figure 2B) algorithms were more suitable for PAAD methylation data compared to the other methods. The color consistency matrix heat map of K = 1 to 10 was obtained according to the consistency clustering of the PAM algorithm (Appendix A) and K-means algorithm (Appendix A).

In the consistency clustering matrix heat map, the values corresponding to the perfect consistency matrix are displayed in blue with white backgrounds on the diagonal blocks on the color-coded heat maps. The results of the PAM and K-means clustering algorithms can be clearly separated into boundaries of visible k colors where the color boundary is relatively clean, showing a white background.

The comparison of the two algorithms showed that for the same k, the PAM algorithm showed a relatively higher consistency coefficient than that of the K-means algorithm; thus, we finally determined the consistency clustering results using the PAM algorithm. The PAM algorithm had the highest average consistency coefficient and stable classification when k = 2 in the clustering process from k = 2 to 10 (Figure 2C). In addition, when k = 2, the corresponding classification color blocks of samples near this value do not often change, and the classification under this value is relatively stable (Figure 2D). Therefore, we used k = 2 as the appropriate number of cluster categories for the next step of the analysis, identifying two methylation subtypes (Cluster1 and Cluster2) with significant prognostic differences for patients with PAAD.

### 3.3. Subtype Analysis of the Prognosis Prediction Model Based on PAAD Classification

Kaplan–Meier survival analysis of the two DNA methylation subtypes was performed using the *Survfit* and *Survdiff* functions in the survival R package. Among 158 samples in the training set, 66 and 62 belonged to subtypes Cluster1 (red) and Cluster2 (blue), respectively, and the analysis results showed a significant difference in prognosis between the two subtypes (*p* = 0.00034). Cluster2 had a better prognosis than that of Cluster1, indicating that the two methylation subtypes of PAAD defined in this study would be useful in predicting the prognosis of PAAD in patients. (Figure 3A).

Next, we further mined DNA methylation markers of the unique methylation patterns between the two subtypes, We first screened 58 differentially methylated genes between the two PAAD subtypes. Furthermore, the corrected *p*-values showed the top 20 genes with the most significant differences (Appendix A). The two PAAD subtypes were identified using consensus clustering, which suggested that samples within both subtypes had unique methylation patterns. Therefore, we further screened for specifically hypermethylated and hypomethylated genes in the two subtypes using QDMR [18], which identified 14 subtype-specific genes in Cluster1 and Cluster2 (Table 1).

We identified 14 subtype-specific genes that were differentially methylated among the two PAAD DNA methylation subtypes, representing their unique methylation patterns (Figure 3B). For Cluster2, associated with a good prognosis, 11 genes showed specific hypermethylation (*CAPN8*, *CSTA*, *GSDMC*, *HEBP1*, *HRH1*, *KRTAP2.4*, *SNORD114.29*, *MAP4K5*, *MIR135B*, *SLAMF7*, and *UCA1*), whereas. The other three genes showed specific hypomethylation (*EVC*, *HOXB4*, and *LOC728392*). Markers of Cluster1 were found to be involved in six main biological functions, including positive regulation of smoothened signaling pathway and somatic stem cell division. Cluster2 markers were mainly involved in 27 biological functions, including MAPK cascade, positive regulation of inositol trisphosphate biosynthetic process, and positive regulation of vascular endothelial cell proliferation (Appendix A).

The results of the model performance evaluation using a 10-fold cross-validation method showed that the constructed model had a classification accuracy of 93.75% (Table 2). In Cluster1, 63 of the 66 samples were correctly classified using the model and for Cluster2, 57 of 62 samples were correctly classified. Moreover, the sensitivity and specificity of the classification model were high. (Figure 3C).

The Bayesian network-based prediction model of PAAD classification was used to analyze 56 samples in the test set (Figure 3D). The 56 samples were assigned labels designating their classification (Cluster1 and Cluster2), and the two classes of samples in the test set also showed similar prognostic profiles to those in the training set. *p*-values indicated statistical significance of survival differences. Furthermore, 30 and 26 samples were classified with the Cluster1 and Cluster2 methylation subtype tags, respectively. In addition, in the prognosis of samples at 1, 3, and 5 years, Cluster2 was superior to Cluster1, and the survival curves were clearly separated without crossover. The log-rank test showed a significant difference in survival curves (*p* = 0.014).

### 3.4. DNA Methylation Subtype Analysis Based on Transcriptome Data

We next focused on the expression differences of several gene sets of interest between two PAAD DNA methylation subtypes. A total of 723 cancer driver genes were obtained from the Catalogue of Somatic Mutations in Cancer (COSMIC)–Cancer Gene Census (CGC) project [22]. Among the DNA methylation subtypes identified in this study, 301 tumor driver genes were differentially expressed between the two groups of samples (Figure 4A).

Of the 78 methylation molecular markers identified in this study that were associated with the prognosis of patients with PAAD, 27 showed significantly different expression levels between the two groups (Figure 4B).

In addition, 4 of the 14 subtype-specific methylated genes identified based on QDMR showed significant differences in RNA expression levels between the two groups (Table 3). Among them, *HRH1* encodes a member of the rhodopsin-like G-protein-coupled receptor family, and previous studies have proven that its expression is associated with poor prognosis [23]. Activation of *SLAM* family member 7 (*SLAMF7*) [24], a self-ligand on T cells, induces phosphorylation of signal transducer and activator of transcription 1 (*STAT1*) and *STAT3*, and inhibits the expression of multiple receptors and transcription factors associated with T-cell exhaustion.

### 3.5. Abundance of Immune Cells in the DNA Methylated Subtypes

ImmuCellAI was used to evaluate the abundance of 24 immune cell types (including 18 T-cell subsets) in PAAD samples based on transcriptome data (Figure 5). The results showed significant differences in the abundance of multiple immune cell types in the PAAD DNA methylation subtypes, Cluster1 and Cluster2 (*p* < 0.05). For example, iTregs, nTregs, Tr1, and monocytes were significantly enriched in Cluster2, which may be related to their immunosuppressive properties. Some antitumor cells such as NK, NK T, and MAIT cells also showed high infiltration in Cluster2, suggesting that the tumor microenvironment may prevent these immune cells from entering Cluster1 tumor nodules. We also found that the infiltration of 17 types of immune cells was significantly different between the two subtypes according to DNA methylation patterns. Consequently, these differences may have the potential to be useful in elucidating the interaction between PAAD and the immune microenvironment. Furthermore, these cells could provide valuable predictors, and the classification model based on DNA methylation patterns for screening the advantage of PAAD immunotherapy is of great significance.

## 4. Discussion

Aberrant DNA methylation is a key factor in the prognostic outcome of cancer, inspired by the study of tumor-typing based on gene expression abnormalities, typing based on the methylation pattern has become a research hotspot that aims to better explain the consistency within the same cancer. Furthermore, this approach may be a further guide in clinical practice to improve the application and value of DNA methylation as a prognostic marker. Zhang et al. [25] identified nine sub-groups in breast cancer based on DNA methylation status. Compared with the PAM50 classification of breast cancer based on gene expression level, DNA methylation subtypes were more refined. Wang et al. [26] identified six subtypes and 17 key hub CpG sites in early stage non-small cell lung cancer (NSCLC). Capper et al. [27] developed a classification method for CNS tumors based on DNA methylation, which identified 91 different types of CNS tumors using methylation features.

The subtypes of PAAD are crucial in determining treatment strategies in clinical care, but there are still no validated staging criteria. Multiple studies have now found the potential feasibility of using DNA methylation data for determining PAAD molecular characteristics based on mining and have explained the potential heterogeneity of PAAD. In an evaluation of 150 patients with PAAD based on TCGA data, DNA methylation data distinguished two major subgroups (H1 and H2). The H1 cluster had more extensive DNA hypermethylation than that of the H2 cluster [28]. Mishra et al. [29] identified three possible subtypes of PAAD based on genome-wide methylation patterns, somatic mutations, copy number changes, and histological features.

In this study, it was proposed for the first time that pancreatic tumors can be divided into two prognostic molecular subtypes, Cluster1 and Cluster2, according to DNA methylation levels, and proved that Cluster2 had a better prognosis than that of Cluster1. Our consistency clustering method based on DNA methylation patterns ensured high intraclass consistency within PAAD molecular subtypes. Furthermore, we further identified 14 specific methylation genes that could represent the unique methylation patterns of the two subtypes. For Cluster2, with a good prognosis, 11 genes showed specific hypermethylation (*CAPN8, CSTA, GSDMC, HEBP1, HRH1, KRTAP2.4, SNORD114.29, MAP4K5, MIR135B, SLAMF7*, and *UCA1*). Three genes (*EVC, HOXB4*, and *LOC728392*) showed specific hypomethylation in Cluster1.

Furthermore, the mined molecular markers of PAAD subtypes were used to construct a prognosis prediction model for PAAD classification based on the Bayesian network. Some of these genes have also been found to be significantly associated with cancer in previous studies. Wuyts et al. [30] demonstrated that *EXTL1* is a member of the *EXT* oncogene family, whereas Redon et al. [31] showed that *SMG6* interacts with telomerase, the chromosome terminal replicase. Inozume et al. [32] found that the *FCRLA* gene is a potential target antigen in immunotherapy of B-cell lymphoma. Whether these genes are associated with the prognosis of PAAD still needs to be further verified through in vivo and in vitro experiments. In addition, genes such as *C11orf34* and *SPATA21*, which ranked among the top 15 most significant, have rarely been reported to be related to tumor development.

In this study, the two PAAD subtypes identified based on DNA methylation differed significantly in terms of survival time, expression levels of cancer driver genes, and abundance of immune cells. In previous studies, it has been reported that a variety of solid tumours can be identified as two distinct tumour subgroups: ‘immune hot’ and ‘immune cold’, which display differing prognosis and oncogenic driver events. This result is consistent with the two classifications obtained in the present study, for the better prognosis Cluster2, which had a higher proportion of oncogenic immune cells enriched and a more intense immune response [33]. At the same time, 14 DNA methylation markers were identified to determine PAAD typing, which provides theoretical support for future precise treatment of PAAD and can improve the correct rate of PAAD treatment and prediction. This strategy may provide a new perspective for the precise treatment of PAAD. Although all of the training data features used in this study are continuous variables, the Bayesian network classifier itself can also accept discrete variables or a mixture of continuous and discrete variables. Such a model would be somewhat more scalable and optimizable as knowledge of PAAD improves.

## 5. Conclusions

In this study, two PAAD methylation subtypes (Cluster1 and Cluster2) were identified based on an unsupervised consistent clustering algorithm, with Cluster2 associated with a better prognosis than Cluster1. Fourteen subtype-specific methylation genes were identified, which could serve as molecular markers to characterize the unique methylation patterns of the two PAAD subtypes. The two DNA methylation subtypes also showed differences in the distribution of clinical features. Transcriptome analysis revealed significant differences in RNA expression levels of many key cancer driver genes, as well as in the abundance of infiltrating immune cells between the two subtypes. The described methodological approach and associated results of this study provide evidence and new directions to support future research on DNA methylation molecular typing as a basis for treatment guidance, prognosis assessment, and further related investigations of other malignant tumors.

## Figures and Tables

**Figure 1 genes-13-01913-f001:**
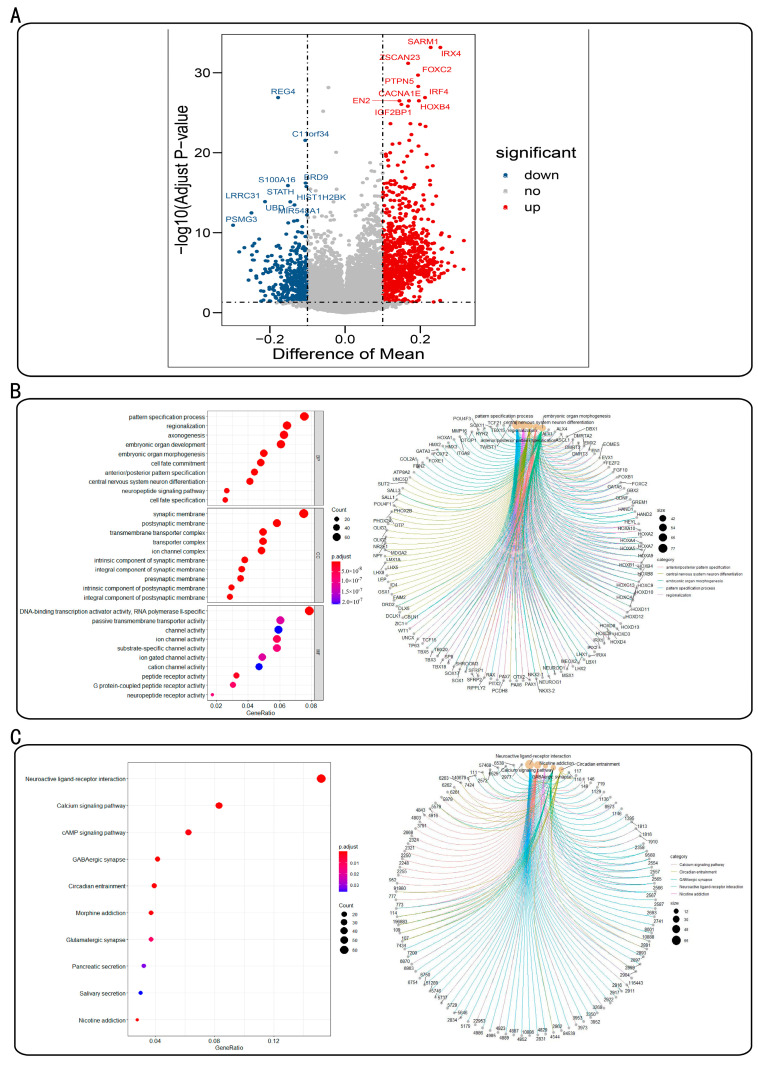
(**A**) Distribution of differentially methylated genes in PAAD. (**B**) Gene ontology (GO) enrichment analysis of differentially methylated genes. Bubble diagram of GO enrichment analysis results and several of the most prominent GO terms associated with the genes. (**C**) Kyoto Encyclopedia of Genes and Genomes (KEGG) pathway enrichment of differentially methylated genes. Bubble diagram of KEGG enrichment and several of the most prominent pathways analysis results for the enriched genes. In the bubble graphs, the horizontal axis is GeneRatio, representing the proportion of genes contained in each enriched function or pathway in the total genes involved in the analysis, and the name of the enriched function or pathway is presented on the vertical axis. The size and color gradient of dots represent the number of genes and *p*-value. Small circles of the network diagram represent the most prominently enriched function or pathway name, distinguished by different colors, and genes in these entries are connected by corresponding color lines.

**Figure 2 genes-13-01913-f002:**
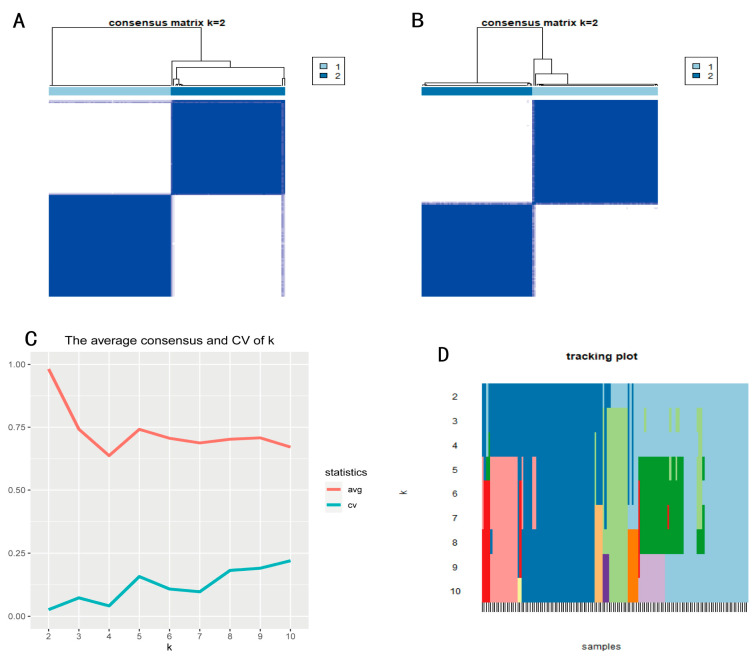
(**A**) Consistency cluster analysis of PAAD samples based on the PAM algorithm. (**B**) Consistency cluster analysis of PAAD samples based on the K-means algorithm. (**C**) Mean coefficient of consistency and coefficient of variation. (**D**) Consistency cluster matrix heat map for k = 2.

**Figure 3 genes-13-01913-f003:**
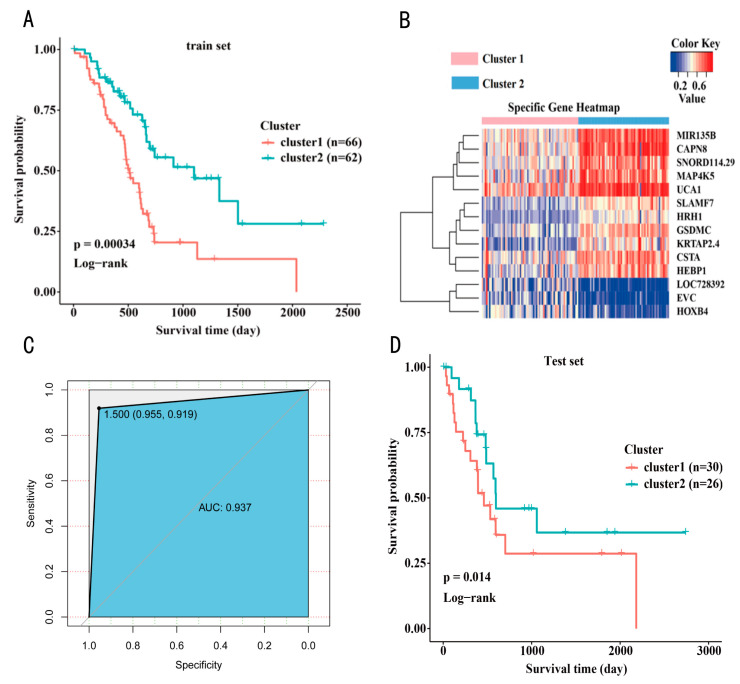
(**A**) Survival curves of two DNA methylated molecular subtypes in the training set. (**B**) Methylation levels of specific genes in two DNA methylation molecular subtypes. (**C**) The specificity and sensitivity of the predictive model. (**D**) The survival curves of the two DNA methylated molecular subtypes in the test set.

**Figure 4 genes-13-01913-f004:**
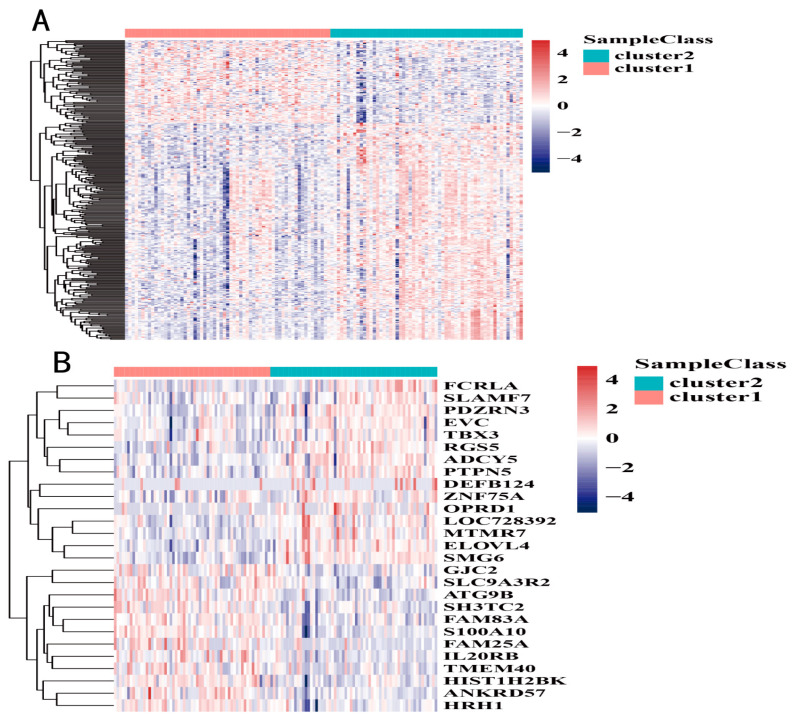
(**A**) Differentially expressed driver genes in two DNA methylation subtypes; (**B**) DNA methylation molecular markers differentially expressed in two DNA methylation molecular subtypes.

**Figure 5 genes-13-01913-f005:**
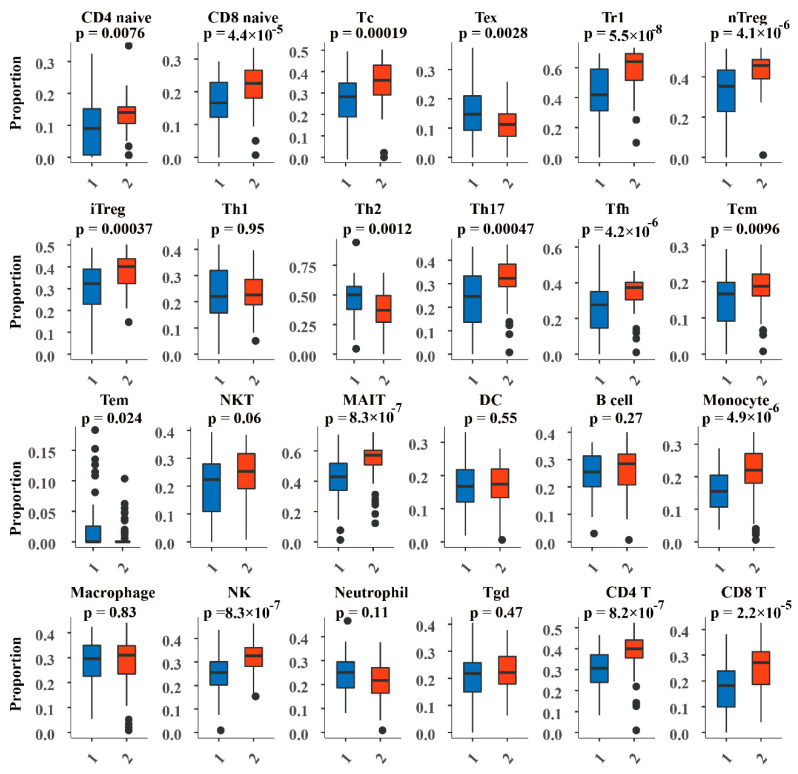
Abundance of 24 immune cells in Cluster1 and Cluster2.

**Table 1 genes-13-01913-t001:** Specific genes associated with PAAD subtypes.

Gene Name	Entropy	Cluster1	Cluster2
*CAPN8*	8.4890	0.4903	0.7586
*MIR135B*	8.6130	0.5052	0.7505
*LOC728392*	8.7880	0.3407	0.1245
*KRTAP2.4*	8.8170	0.3170	0.5288
*GSDMC*	8.8670	0.3958	0.6001
*HEBP1*	8.8730	0.4158	0.6190
*SNORD114.29*	8.9020	0.4707	0.6698
*MAP4K5*	8.9140	0.4968	0.6941
*EVC*	8.9250	0.3252	0.1295
*UCA1*	8.9460	0.5976	0.7903
*CSTA*	9.0130	0.4692	0.6528
*SLAMF7*	9.0760	0.3656	0.5409
*HOXB4*	9.1070	0.3848	0.2134
*HRH1*	9.1590	0.3117	0.4766

GeneName: gene symbol; entropy: Shannon entropy of differentially methylated genes; Cluster1 and Cluster2: mean methylation level of specific genes in subtype 1 and subtype 2, respectively.

**Table 2 genes-13-01913-t002:** Classifier confusion matrix based on the Bayesian network.

	Prediction Cluster1	Prediction Cluster2
Cluster1	63	3
Cluster2	5	57

**Table 3 genes-13-01913-t003:** Subtype-specific methylated genes with significantly different RNA expression levels.

Gene	Fold Change	Adjusted *p*-Value
HRH1	2.3112	$1.93\times 10^{-5}$
LOC728392	0.3919	$1.49\times 10^{-4}$
SLAMF7	0.3007	$1.09\times 10^{-2}$
EVC	0.3859	$1.34\times 10^{-2}$

## Data Availability

No new data were created or analyzed in this study. Data sharing is not applicable to this article.

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
