# Peer review of "Classification and Prognosis Analysis of Pancreatic Cancer Based on DNA Methylation Profile and Clinical Information"

_genes, 2022, doi:10.3390/genes13101913_

Round 1

Reviewer 1 Report

I have checked the manuscript (title: Classification and Prognosis Analysis of Pancreatic Cancer Based on DNA Methylation Profile and Clinical Information). I enumerate some comments as follows.  

Major points

1.     Could you identify characteristic pathway or function in methylated genes in cluster1 and cluster2? If characteristic patterns in two subtypes were identified, authors should show them.

2.     In both the train set (Figure 7) and test set (Figure 10), survival probability was significant higher in cluster2 than in cluster1. However, the p value was considerably different. Authors should indicate why the difference has happened.

Minor point

1.     Are the positions of genes ‘LOC728392’ and ‘SNORD114.29’ opposite in the page 11, line 1 and 3 from the bottom?

2.     The paragraph ‘2.’ is lost in session 3.3.1 in the page 14.

3.     In Table 7, the titles ‘Gene Name’ are duplicated.

Reviewer 2 Report

Abstract 

Please clarify what data was obtained from SEER – this sentence is confusing as the methylation data did not come from SEER:  “We used consistency clustering to explore PC-specific methylation patterns, using information from the Surveillance, Epidemiology, and End Results (SEER) database.” 

The abstract needs improvement. It is unclear whether the “PC risk factors” are known risk factors associated with survival or incidence; it would be helpful to specify what these are. Also, provide the number of PC cases (cancer and normal tissue) and number of outcomes for prognosis (death, progression?)

“Combining RNA expression and clinical data from the SEER database,” Specify that RNA expression is from TCGA and specify which clinical data are included here. 

Introduction. Last paragraph: clarify what data was obtained from TCGA vs SEER databases. 

Methods.

The methods are unorganized, and it is hard to follow the steps that were taken to conduct the study. It would be better to start the Methods providing detail information about the source of the dataset and what variables were available from each of these. For example, it is unclear if the TCGA dataset provided information on prognosis. SEER is mentioned in the abstract and introduction but not in the Methods. What data was obtained from SEER? and how were they merged with the TCGA dataset given the SampleId would not be the same in a SEER dataset. Please specify numbers of tumors, adjacent, patients with stage information, etc. details for these are missing. Was time from diagnosis to death also available? (Presumably since Kaplan-Meier curves are mentioned, but this needs to be clarified). 

After providing the details on TCGA data and SEER data, the specific information on methylation and RNA expression processing can be given.

Finally, there should be a statistical analysis section, describing in detail how the analysis was conducted (clustering of data, etc) and how these are modeled for progression with other covariates. Were covariates such as age, sex, race/ethnicity, and treatment available and were these considered in the prediction model? 

Provide more details for this: “The training set was used for model construction and screening of prognostic markers” 

“Pancreatic Cancer Subtype Recognition” – specify what “subtypes” are used for this section.

Results: Some of these descriptions belong in the Methods. E.g. “After the preprocessing steps, data containing 131288 CpG site methylation values of each sample were obtained” Also, please explain how the number of sites were reduced from ~480K to ~ 131K.  This is a much greater removal of probes than expected from pre-processing. 

In the screening of genes (Section 3.1.2) were immune cell types adjusted for (this can be done for DNA methylation dataset). 

In Section 3.1.3 the analysis using Cox Proportional Hazard model (not mentioned in the Methods). Please describe in the Methods how this was model (what is the person time used?) 

“In addition, the univariate COX proportional hazards regression model established for clinical factors (Table 2) showed that "age" (P = 0.021) was also a significant factor affecting the survival of patients.” If age is in the model (and clinical factors? Which ones?) the model is not univariate. If you mean that Table 2 is the analysis one methylation site at a time, this needs to be clarified.  The title of the table should specify what the hazard ratio is for (lung cancer deaths). Also specify the increment for the methylation level. Please replace CI95 with 95% confidence interval. 

Given that genes can have many CpG sites, it would be better (and more conventional) to provide the Cg number. 

Figures 4 and 5 should be supplemental data. 

Figure 7 should be the results for the test set, not the training set. 

Figure 9 shows that the results identified are driven by immune cell differences; these should have been adjusted for in the models that derived the clusters.  Immune cells have differentially methylated regions (DMRs) that are unique to each one and not adjusting for immune cell types in the analysis is a major problem.

Reviewer 3 Report

Lin et al submitted a very interesting topic discussing the specific DNA methylation patterns for 184 samples of patients with pancreatic cancer. The topic is very interesting to the readers and authors have a good quality of presentation. Overall, the manuscript was well-written and organized; I have, however, four points that might help authors to make the content more scientific and significant.

(1)    The last paragraph of the introduction doesn’t have a clear statement for the aim of the study. This is very important to attract reader.

(2)    High resolution figures are very important for publication, so I would suggest using 300 or 600 DPI for figures 4 and 5. Also, the information in figure 6a should be in English language.

(3)    All abbreviations in figure 13 should be written in a full name in the legend.

(4)    The conclusion should be rewritten to include the main findings rather than writing the way of doing your experiments.

I I wish the authors the best of luck with their revision process.

Round 2

Reviewer 1 Report

I have re-checked the manuscript (title: Classification and Prognosis Analysis of Pancreatic Cancer Based on DNA Methylation Profile and Clinical Information). Authors have faithfully corrected the points which I pointed out. I have no points which I require further correction.

Author Response

We really thank so many useful comments and suggestions from the reviewer.

Reviewer 2 Report

The authors did not address the comment made in the prior review about the adjustment of immune cell types in the analysis that uses DNA methylation to develop the predictive markers. The article by Chakravarthy, A., Furness, A., Joshi, K. et al. Pan-cancer deconvolution of tumour composition using DNA methylation. Nat Commun 9, 3220 (2018). https://doi.org/10.1038/s41467-018-05570-1 provides the information necessary to adjust for different immune cell types in the tissue using DNA methylation data. Not considering the mixed cell type combinations in the analysis could lead to results that are not replicable. If the authors do not make the adjustment, this should be discussed as a limitation to the study findings. The two clusters identified could simply represent different composition of immune cell types in the pancreatic cancer tumors. While this information itself is useful, it is important to point out that the markers identified may simply be differentially methylated regions (DMRs) for specific subsets of immune cells. Given the differences in immune cell proportions for the 2 clusters (as shown in Figure 5), there is high likelihood that it is in fact what is happening with these findings. 

Author Response

请参阅附件。

Reviewer 3 Report

The manuscript has extensively been improved. Authors have properly responded to all suggested points. 

Author Response

We really thank so many useful comments and suggestions from the reviewer.The English language in the revised manuscript has been carefully corrected to improve grammar and readability.

Round 3

Reviewer 2 Report

The clusters are indeed likely to represent different type of tumors based on immunological components. I was hoping the authors would note the possibility that the DNA methylation regions identified are precisely measuring that (given that immune cells, by lineage, can be identified with changes in DNA methylation), rather than representing oncogenic pathways.